# Scaling Behavior in Model Fine-tuning for Audio DeepFake Detection

**Xiang Li** [1]  **Pin-Yu Chen** [2]  **Wenqi Wei** [1]

## Abstract

Recent advances in audio deepfake detection have been driven by increasingly large speech foundation models and growing amounts of synthetic data. Despite strong benchmark performance, it remains unclear how detection capability scales with model capacity and training data under realistic deployment conditions involving distribution shift, signal corruption, and unseen synthesis pipelines. In this work, we present the first systematic study of scaling laws in post-training audio deepfake detection, focusing on fine-tuning regimes rather than large-scale pretraining. Using a controlled family of speech foundation models with shared architecture and pretraining, we analyze how detection performance, robustness, and generalization evolve as a function of model size and training data scale. Our results reveal a fundamental asymmetry between performance scaling and robustness scaling in audio deepfake detection, suggesting increasing model capacity alone is insufficient for achieving reliable real-world generalization.

## 1. Introduction

Audio deepfake detection has become a critical component in safeguarding digital communication, as recent advances in Text-to-Speech (TTS) and Voice Clone (VC) have enabled the generation of highly realistic and scalable fake audio. Recent audio deepfake detectors increasingly rely on large self-supervised speech foundation models, which have demonstrated strong performance across multiple benchmarks. As both model capacity and training data continue to grow, a natural question arises: how does detection performance scale, and under what conditions does scaling meaningfully improve robustness and generalization?

While scaling laws have been extensively studied in language modeling and representation learning, their implications for post-training audio deepfake detection remain unclear. Existing studies (Wang & Yamagishi, 2021; Kawa et al., 2023; Li et al., 2025b; Ge et al., 2025) using speech foundation models as detectors mainly report benchmark results or compare a small number of model variants, often under clean conditions. However, real-world deployment presents a far more challenging landscape, involving unseen synthesis methods, distribution shifts across datasets, acoustic corruptions, and cross-lingual inputs. Whether improvements from scaling persist or saturate under these conditions is still unexplored.

In this work, we conduct a comprehensive empirical study of scaling behavior in audio deepfake detection, with a particular focus on fine-tuning regimes rather than large-scale pretraining. We adopt the Whisper model family (Radford et al., 2023) as a controlled testbed, enabling us to isolate the effects of model capacity while holding architecture and pretraining data constant. By systematically varying both model size and training data scale, we examine how detection performance evolves across a broad range of evaluation settings.

Importantly, our analysis goes beyond in-distribution detection performance. We explicitly study scaling behavior under (i) out-of-distribution benchmarks with unseen synthesis pipelines and recording conditions, (ii) robustness to common audio corruptions such as noise, pitch manipulation, and neural codec distortion, (iii) cross-language generalization to 46 non-English languages despite English-only training, and (iv) cross-TTS generalization to unseen speech synthesis systems. This multi-axis evaluation allows us to characterize not only whether scaling helps, but when and why it fails.

Our findings reveal that while larger detectors are consistently more sample-efficient and achieve lower error rates, scaling behavior differs conspicuously across evaluation axes. In-distribution performance follows clean and predictable power-law trends, whereas robustness to corruptions and linguistic shift exhibits weaker or saturating scaling. These results suggest that scaling alone is insufficient to guarantee robustness and that post-training detector improvements are constrained by representational mismatch

---

[1]Department of Computer and Information Science, Fordham University, New York, USA [2]IBM Research, New York, USA. Correspondence to: Xiang Li <xl5@fordham.edu>.

*Proceedings of the 43rd International Conference on Machine Learning*, Seoul, South Korea. PMLR 306, 2026. Copyright 2026 by the author(s).

rather than data scarcity alone. We summarize our contributions and key experimental insights as follows:

- **Post-training scaling laws for audio deepfake detection**. We provide the first systematic analysis of how detection performance scales with both model capacity and fine-tuning data size in audio deepfake detection. Our results show that larger detectors are consistently more sample-efficient and achieve lower error rates, with performance following stable power-law trends under in-distribution evaluation.

- **A unified view of scaling under robustness and distribution shift**. Through extensive evaluation on out-of-distribution benchmarks and common audio corruptions, we demonstrate a clear asymmetry between detection performance scaling and robustness scaling. While robustness to Gaussian noise improves predictably with capacity, robustness to distortions, such as pitch shifting and neural codec artifacts, exhibits weak or saturating scaling, revealing fundamental limitations of capacity-driven robustness.

- **Scaling behavior under cross-language and cross-TTS generalization**. We show that detector scaling induces emergent generalization to unseen languages and unseen TTS systems, even when training is restricted to English-only data and a limited set of generators. However, cross-language and cross-TTS evaluations exhibit slower scaling rates and persistent error gaps, indicating that scaling alone does not eliminate linguistic or synthesis-specific mismatches.

## 2. Related Work

### 2.1. Scaling Laws for Large Language Models

Scaling laws have emerged as a unifying framework for understanding how model performance improves as a function of scale, including model parameters, dataset size, and training compute. Early empirical studies (Kaplan et al., 2020) demonstrate that language model loss follows a predictable power-law relationship with respect to scale over several orders of magnitude. In general, scaling behavior can be expressed as

$$\mathcal{L}(x) = \alpha x^{-\beta} + \epsilon, \tag{1}$$

where $\mathcal{L}$ denotes the validation loss (or error), $x$ represents a scaling variable such as model size, dataset size, or compute, $\beta$ is the scaling exponent, and $\epsilon$ captures irreducible error.

Subsequent work refines these observations by jointly modeling data scale and model capacity, showing that optimal performance is achieved when model size and dataset size are properly balanced under a fixed compute budget (Hoffmann et al., 2022). More recent works extend

scaling-law analysis beyond pretraining to the post-training stage. Hernandez et al. (2021) shows that downstream performance after fine-tuning follows predictable power-law behavior with respect to model size and fine-tuning data. Zhang et al. (2024a) further demonstrates that fine-tuning performance exhibits joint scaling with model capacity, dataset size, and adaptation strategy, including both full fine-tuning and parameter-efficient methods such as LoRA. However, these works primarily focus on clean downstream performance, leaving the scaling behavior of robustness and generalization under realistic distribution shifts unexplored.

### 2.2. Audio Deepfake Detection

Recent advancements in TTS technology have significantly enhanced the ability to generate high-quality and realistic audio, calling for an urgent need for more robust and reliable detection methods. Previous works have focused on distinguishing AI-generated audio from genuine audio by designing advanced model architectures (Jung et al., 2022; Tak et al., 2021a;b) to extract different levels of representations of speech data for audio deepfake detection. More recent methods leverage self-supervised speech foundation models, including Wav2Vec2.0 (Wang & Yamagishi, 2021), HuBERT (Wang & Yamagishi, 2021), Whisper (Kawa et al., 2023), XLS-R (Zhang et al., 2024b), and Wav2Vec-BERT (Li et al., 2025b), which provide rich and transferable representations for downstream detection tasks.

Several studies have examined the generalization and robustness of such foundation-model-based detectors, showing that larger pretrained models tend to achieve improved performance and robustness across datasets and perturbations (Li et al., 2025b;a). In addition, Ge et al. (2025) demonstrates that post-training speech foundation models on a diverse collection of deepfake datasets can yield substantial performance gains. However, they fail to systematically characterize how detection performance scales with model capacity and training data. To the best of our knowledge, our work is the first to study the scaling behavior of audio deepfake detection, explicitly analyzing how performance, robustness, and generalization evolve as a function of both model size and dataset scale.

## 3. Designing Audio Deepfake Detection Fine-Tuning Scaling Laws

Our goal is to systematically investigate how fine-tuning performance for audio deepfake detection scales with model capacity and training data size under a controlled post-training setting. Rather than studying pretraining effects, we focus exclusively on the fine-tuning stage, which more directly reflects practical detector deployment. As illustrated in **Figure 1**, we fine-tune detectors by jointly varying model size and data scale while holding model architecture and

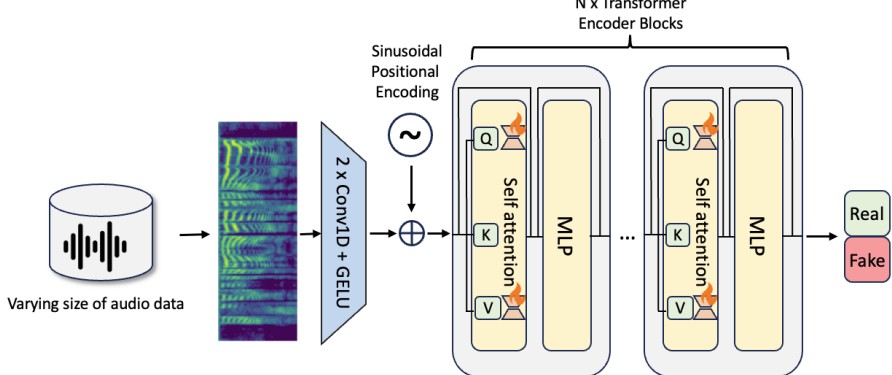

*Figure 1.* Fine-tuning pipeline for Whisper-based detectors. We vary dataset size and model size for finetuning.

pretraining fixed.

**Scaling Formulation.** We model detection performance using power-law scaling with respect to model capacity and training data size (Equation 1), following prior work on neural scaling laws (Kaplan et al., 2020). Unless otherwise stated, scaling curves are fitted using equal error rate (EER) as a function of model size or dataset size under fixed training protocols.

**Models.** To focus on the effect of post-training data scale and isolate the impact of model capacity, we control for both architecture and pretraining data across all experiments. Therefore, we adopt the Whisper model family (Radford et al., 2023) as our foundation model, which shares a common architecture and pretraining corpus while offering multiple model sizes. This design allows us to vary model capacity over more than an order of magnitude (i.e., Tiny (39M), Base (74M), Small (244M), Medium (769M), and Large (1.55B)) without introducing additional factors such as architectural changes or differences in pretraining data. As a result, performance variations observed across models can be more confidently attributed to scaling effects rather than implementation or pretraining data discrepancies. All models are fine-tuned using identical training protocols, enabling a controlled and systematic analysis of how detection performance scales with model size.

## 4. Experiment and Evaluation Setups

### 4.1. Experiemental Settings

**Training Datasets.** We train our models on a diverse collection of publicly available audio deepfake datasets, including ASVSpoof2019 (Wang et al., 2020), ASV5 (Wang et al., 2024), CodecFake (Wu et al., 2024), LibriTTS-train-clean-360 (Zen et al., 2019), DFADD (Du et al., 2024), LJSpeech (Ito & Johnson, 2017), WaveFake (Frank & Schönherr, 2021), and CD-ADD (Li et al., 2024). To iso-

late the effect of *cross-language generalization*, we restrict training to English-only audio and exclude all non-English subsets from the training data. All models are therefore fine-tuned exclusively on English speech, resulting in a training corpus of over 2 million audio samples.

**Evaluation Datasets.** We aim to evaluate scaling behavior under both in-distribution and out-of-distribution (OOD) conditions. For *in-distribution* evaluation, we use the official evaluation splits provided by the training datasets, where test samples share similar synthesis pipelines and recording conditions with the training data. To assess *out-of-distribution* (OOD) generalization, we evaluate on a broad set of unseen and more challenging benchmarks, including In-the-Wild (Müller et al., 2022), ADD2022 (Tracks 1 and 3) (Yi et al., 2022), ADD2023 (Rounds 1 and 2) (Yi et al., 2023), ASVspoof2021 LA (Yamagishi et al., 2021), ASVspoof2021 DF (Yamagishi et al., 2021), Fake-or-Real (Reimao & Tzerpos, 2019), CodecFake (Xie et al., 2025), SONAR (Li et al., 2025b), and LibriSeVoc (Sun et al., 2023). These datasets differ substantially from the training data in terms of synthesis models, codecs, recording environments, and content, enabling a comprehensive evaluation of robustness and generalization.

**Training Details.** We adopt Low-Rank Adaptation (LoRA) (Hu et al., 2022) for parameter-efficient fine-tuning of the detector. Specifically, low-rank adapters with rank $r = 16$ and scaling factor $\alpha = 32$ are injected into the query and value projection layers, with a dropout rate of 0.1 applied to the LoRA modules. All audio is resampled to 16 kHz, volume-normalized, and truncated to a maximum duration of 4 seconds. Models are trained for 5 epochs using the Adam optimizer with a learning rate of $2 \times 10^{-4}$ and a weight decay of $5 \times 10^{-4}$. We employ a constant learning-rate schedule with 500 warm-up steps throughout training.

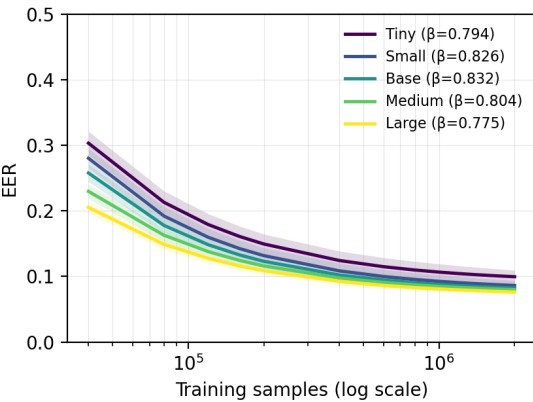

*Figure 2.* In-distribution scaling behavior. Equal error rate (EER) as a function of training set size for detectors of increasing model capacity, evaluated under in-distribution conditions. Performance improves smoothly with additional data for all model sizes, with larger detectors consistently achieving lower EER and exhibiting higher sample efficiency across the entire data regime.

## 5. Results

### 5.1. Scaling with Training Data and Model Size

**TAKEAWAYS: In-Distribution Scaling**

(1) Detection performance follows stable power-law scaling with both model size and training data.
(2) Larger models are consistently more sample-efficient, achieving lower EER with fewer training samples.
(3) Scaling benefits saturate at large data scales, indicating diminishing returns from data alone.

We first examine how audio deepfake detection performance scales jointly with training data size and detector capacity under *in-distribution* conditions. To study data scaling in a controlled manner, we construct a series of training subsets by uniformly subsampling the full training corpus at different fractions: [2%, 4%, 6%, 8%, 10%, 20%, 30%, 40%, 50%, 60%, 70%, 80%, 90% 100%]. Since the training corpus consists of multiple datasets, each subset is created by independently sampling the same fraction from *each* dataset (e.g., 2% corresponds to retaining 2% of samples from every dataset), rather than subsampling from the aggregated pool. This strategy preserves the original dataset composition across scales and ensures that observed performance differences primarily reflect the effect of training data size rather than changes in data distribution or dataset imbalance. All models are trained and evaluated under identical protocols across subsets, enabling a fair comparison across both data scale and model capacity.

**Figure 2** illustrates how in-distribution (ID) detection performance scales jointly with training data size and model capacity. Across all model variants, increasing the amount of training data consistently reduces EER. This monotonic

improvement indicates that detector performance is not bottlenecked by optimization instability or overfitting at small scales, but instead benefits directly from additional labeled data.

A key observation from the in-distribution results is that larger detectors consistently outperform smaller ones across the entire data regime, including at very small training fractions. This indicates that increased model capacity provides a systematic advantage rather than becoming effective only in high-data settings, likely due to the ability of larger models to learn richer and more discriminative representations. Notably, larger models reach a given EER using substantially fewer training samples, demonstrating improved sample efficiency with scale and suggesting practical benefits of higher-capacity detectors even when labeled data is limited.

As training data increases, performance improves smoothly for all model sizes but exhibits diminishing returns at larger scales. The gap between models gradually narrows, indicating partial saturation: while additional data continues to reduce error, marginal gains decrease and scaling curves begin to converge. This behavior is consistent with power-law scaling, where improvements taper as data grows, and suggests that beyond a certain regime, further gains may require increased model capacity or architectural advances rather than simply more data.

Overall, in-distribution detection performance follows stable and predictable scaling trends with respect to both training data and model size. Larger detectors are uniformly more sample-efficient and achieve lower error rates across all regimes, although their advantage diminishes at high data scales due to saturation. These results confirm that post-training audio deepfake detectors exhibit scaling behavior analogous to that observed in other learning domains.

### 5.2. Out-of-Distribution Scaling Behavior

**TAKEAWAYS: Out-of-Distribution Scaling**

(1) Scaling improves OOD performance, but at a slower and less predictable rate than in-distribution.
(2) Performance gaps between model sizes persist under distribution shift, even with large training data.
(3) Data and model size scaling alone cannot fully compensate for distribution shift in OOD settings.

We next analyze scaling behavior under out-of-distribution (OOD) conditions, where test data differs from the training distribution in synthesis methods and acoustic characteristics. As presented in **Figure 3**, although increasing training data and model size continue to improve detection performance, the gains are systematically weaker and more variable than those observed in in-distribution settings. This reduced slope indicates that OOD generalization is limited primarily by distribution mismatch rather than by data quan-

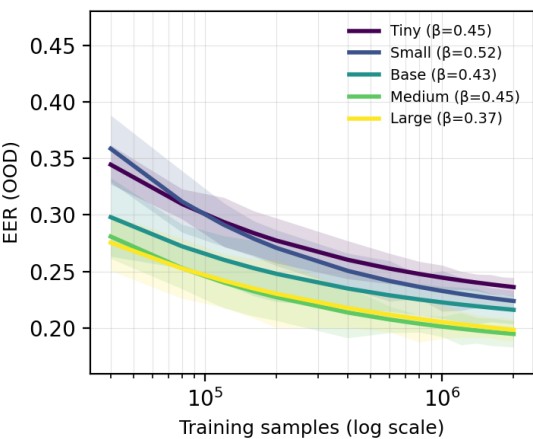

*Figure 3.* Out-of-distribution scaling behavior. While performance improves with additional training data and larger models, gains are consistently weaker than in-distribution scaling and exhibit higher variance. Larger detectors still exhibit better detection performance, but performance gaps persist at large data scales, indicating that scaling alone does not fully resolve generalization under distribution shift.

tity alone, and that additional in-distribution data yields diminishing returns for unseen conditions.

However, the relative ordering of model sizes is largely preserved: larger detectors generally outperform smaller ones across data scales. However, unlike in-distribution evaluation, performance gaps do not close at larger data sizes, revealing a persistent generalization error under distribution shift. Notably, the scaling curves of the Medium and Large models become nearly indistinguishable. This convergence implies that in our case, beyond a certain model size, further capacity increases offer limited advantage without corresponding changes in training diversity or objectives.

At smaller scales, we observe a crossover between the Tiny and Small models, indicating that increased capacity does not uniformly translate into better OOD performance. This behavior suggests that when capacity is limited, training dynamics and inductive biases may interact with distribution shift in non-monotonic ways, leading to regime-dependent advantages. Such crossovers further underscore that OOD scaling is less stable and predictable than in-distribution scaling.

Overall, while OOD performance benefits from increased data and model size, scaling alone does not eliminate generalization gaps under distribution shift. Larger models retain an advantage, but capacity gains saturate and do not fully address OOD failure modes. These findings indicate that reliable OOD audio deepfake detection requires complementary strategies—such as diversity-aware training, targeted augmentation, or robustness-oriented objectives—beyond simply increasing scale.

## 5.3. Robustness to Audio Corruptions

**TAKEAWAYS: Robustness to Audio Corruptions**

(1) Robustness to Gaussian noise scales similarly to clean EER.
(2) Robustness to structured distortions (pitch shift, Encodec) exhibits weak or saturating scaling.
(3) Model scaling alone does not guarantee robustness under realistic audio perturbations.

Following prior work (Li et al., 2025a), we investigate how robustness against common audio corruptions scales with detector capacity. Robustness against perturbations poses a distinct challenge: corruptions may preserve semantic content while selectively altering acoustic cues exploited by detectors. Understanding whether such robustness emerges naturally through model scaling is therefore critical for reliable deployment.

We evaluate detection performance under three representative corruptions from (Li et al., 2025a): Gaussian noise, pitch shifting, and neural codec distortion (Encodec). For each corruption, we measure EER as a function of model size and fit power-law curves to both clean and corrupted conditions, enabling a direct comparison of robustness scaling behavior.

Under Gaussian noise, EER decreases monotonically with model size, closely mirroring the clean-condition scaling curves. The corrupted curves exhibit similar slopes and remain approximately parallel to the clean curves across all model sizes, with only a small and consistent vertical offset. This behavior indicates that Gaussian noise primarily induces a constant degradation in performance rather than altering the underlying scaling law. Robustness to unstructured noise, therefore, improves predictably with model capacity, and scaling remains an effective means of reducing error under this corruption.

For pitch shifting, detection performance still improves with increasing model size; however, the scaling curves under corruption are noticeably flatter than their clean counterparts. The gap between clean and pitch-shifted performance persists across all scales, indicating weaker robustness gains from additional capacity, which indicates that while larger models reduce absolute error, scaling does not substantially mitigate sensitivity to pitch perturbations. Unlike Gaussian noise, pitch shifting alters structured acoustic properties, leading to diminishing returns from model scaling.

Neural codec distortion exhibits the weakest scaling behavior. Although clean-condition EER continues to decrease with model size, performance under Encodec distortion improves only marginally, resulting in a large and persistent gap between clean and corrupted curves.

The corrupted scaling curve shows signs of saturation at

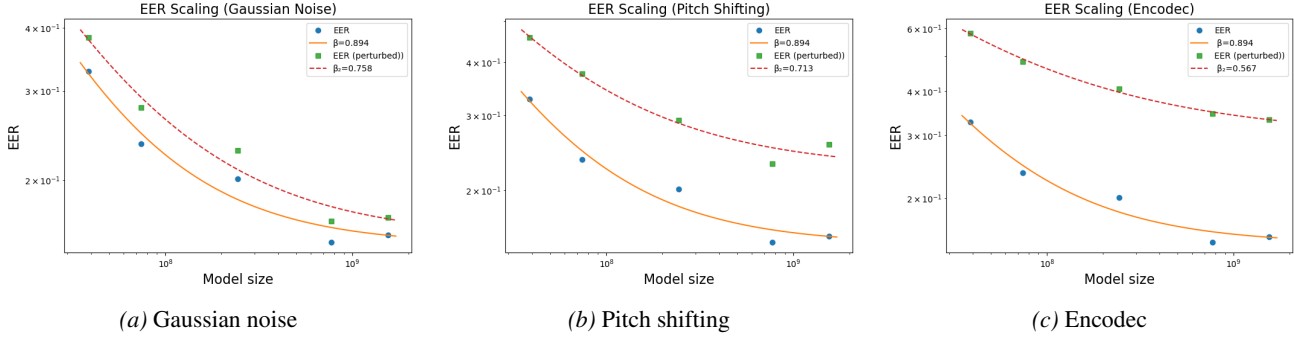

*(a)* Gaussian noise      *(b)* Pitch shifting      *(c)* Encodec

*Figure 4.* EER as a function of model size under three representative audio corruptions: (a) Gaussian noise, (b) pitch shifting, and (c) neural codec distortion (Encodec). Solid curves show clean-condition scaling, while dashed curves correspond to corrupted evaluation. Robustness improves with increasing model capacity across all corruptions, but the rate of improvement varies substantially. Under Gaussian noise, robustness closely follows clean-condition scaling, indicating that noise primarily induces a uniform performance shift. In contrast, pitch shifting and Encodec distortions exhibit weaker scaling behavior and larger persistent error gaps.

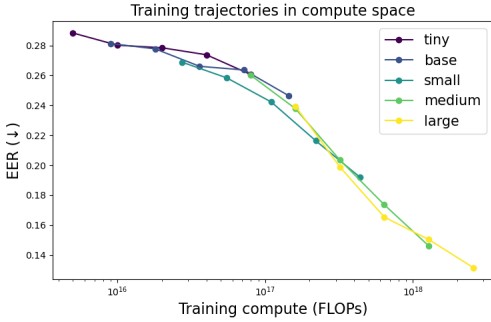

*Figure 5.* Compute–performance trade-offs. EER versus training compute for detectors of increasing capacity. Performance improves with compute, and intersections among training trajectories indicate compute-dependent optimal model sizes rather than a single model dominating across all budgets.

larger model sizes, suggesting that increasing capacity yields limited additional robustness. This indicates that robustness to neural codec artifacts does not follow the same scaling behavior as the clean conditions.

## 5.4. Compute–Optimal Training Regimes

> **TAKEAWAYS: COmpute-Optimal Regimes**
>
> Optimal detector performance is compute-dependent: smaller models are preferable under limited budgets, while larger models benefit only when trained with sufficient compute.

We further study how detection performance scales with training compute under fixed pretraining, and whether larger detector models are compute-efficient across the finetuning regime. **Figure 5** shows training trajectories of Whisper-based detectors of increasing capacity, where each curve corresponds to a fixed model size and reports EER as a function of total compute during finetuning.

Across all model sizes, detection performance improves

monotonically with increasing compute, exhibiting clear diminishing returns at higher budgets. At low compute levels, smaller models achieve lower EER than larger counterparts, indicating that high-capacity detectors are initially undertrained and fail to fully utilize their representational capacity. As compute increases, medium and large models improve more rapidly and eventually surpass smaller models, achieving substantially lower error at higher budgets.

Crucially, the training trajectories intersect at intermediate compute levels, demonstrating that the compute-optimal model size depends on the available training budget. No single model dominates across the entire compute range. Instead, limited compute is most effectively allocated to smaller models that can be trained closer to convergence, whereas larger budgets favor higher-capacity models trained for longer durations. This behavior reveals a fundamental trade-off between model capacity and optimization depth under fixed compute constraints.

These results indicate that post-training detector scaling is governed by compute allocation rather than model size alone. Increasing model capacity without sufficient training compute leads to suboptimal performance, while appropriately matching model size to available compute yields more efficient use of resources. Together, these observations motivate a compute-optimal perspective on detector scaling, in which model capacity and training steps must be jointly considered to achieve optimal performance under resource constraints.

## 5.5. Cross-Language Generalization

To examine whether detector scaling generalizes beyond the training language, we evaluate models trained exclusively on English data on the SpeechFake benchmark (Huang et al., 2025), which contains synthesized speech from 46 non-English languages spanning diverse phonetic inventories and prosodic patterns. **Figure 6** reports EER as a function

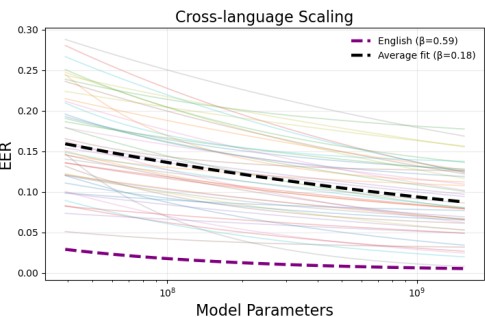

*Figure 6.* Cross-language scaling. Solid curves show EER as a function of model size for individual unseen languages. The dashed black curve represents the average performance across languages, while the dashed purple curve shows in-distribution English performance. Detection performance improves consistently with increasing model capacity across languages, but there are substantial language-specific performance.

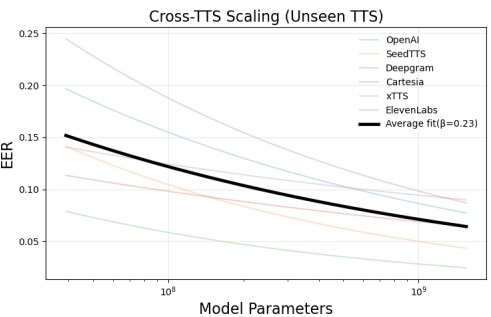

*Figure 7.* Cross-TTS scaling. Each curve corresponds to a different generator, while the bold black line shows the average trend across generators. While EER decreases consistently with increasing model capacity, generator-specific gaps remains and are not eliminated by scaling alone.

of model size for each language, together with an averaged scaling trend across all non-English languages.

> **TAKEAWAYS: Cross-Language Scaling**
>
> (1) Detector scaling induces emergent generalization to unseen languages.
> (2) Cross-language scaling is weaker than in-distribution scaling.
> (3) Language-specific performance gaps persist across model sizes.

Across languages, detection performance consistently improves as model capacity increases. Despite large variation in absolute EER across languages, the majority of language-specific curves exhibit a similar monotonic decrease with scale, indicating that scaling detector capacity induces systematic cross-language generalization, even in the absence of multilingual training data.

However, the improvement rate under cross-language evaluation is notably weaker than in in-distribution settings. The averaged scaling trend exhibits a substantially smaller exponent than that observed for English, reflecting slower error reduction as model size grows. Moreover, the relative performance gaps between languages persist across all scales: languages with higher EER at small model sizes remain comparatively harder even for the largest detectors. This suggests that scaling alone does not eliminate language-specific mismatches arising from phonetic, prosodic, or acoustic differences.

Importantly, the consistency of the scaling trends across languages indicates that increased model capacity does not overfit to English-specific cues, but instead improves the extraction of more universal acoustic features relevant to deepfake detection. At the same time, the persistent offsets across languages imply that such universal representations are insufficient to fully bridge cross-lingual gaps without explicit multilingual supervision.

## 5.6. Cross-TTS Generalization

> **TAKEAWAYS: Cross-TTS Scaling**
>
> (1) Scaling improves detection on unseen TTS systems in a consistent and predictable manner.
> (2) Performance improvements are driven primarily by model capacity rather than generator-specific effects.
> (3) A persistent baseline gap remains, indicating incomplete synthesis-invariant generalization.

In this section, we investigate whether detector scaling generalizes across unseen speech synthesis systems. While previous analyses focus on distribution shifts induced by language or acoustic corruption, cross-TTS evaluation probes generalization to differences in speech synthesis pipelines, including vocoders, acoustic models, and synthesis artifacts. We evaluate detectors trained without exposure to a given TTS system on audio generated by that unseen system, and analyze how performance scales with model capacity.

**Figure 7** reports EER as a function of model size for several unseen TTS generators, along with the averaged scaling trend across all systems. This setting isolates generalization across synthesis mechanisms rather than linguistic or signal-level variation.

Across all unseen TTS systems, detection performance improves steadily with increasing model capacity. Although EER varies substantially across different TTS systems, which reflects the differences in synthesis quality and artifact characteristics, the scaling trends remain consistent, with a monotonic decrease as the model size increases. Notably, the relative ordering of TTS systems remains largely preserved across scales: generators that are harder to detect at small model sizes remain comparatively harder even for larger detectors. This indicates that increasing detector capacity systematically improves generalization to previously

unseen synthesis methods.

Compared to in-distribution evaluation, the cross-TTS scaling curve exhibits a higher baseline error, indicating a persistent generalization gap. However, the absence of curve saturation within the evaluated range suggests that increasing capacity may continue to yield meaningful gains under cross-TTS conditions.

## 6. Discussion

**Implications.** Our study provides a unified view of how post-training scaling affects audio deepfake detection across detection performance, robustness, and generalization. A key takeaway is that scaling yields consistent but non-uniform benefits across evaluation axes. In-distribution performance follows clean and predictable scaling trends, while generalization under distribution shift—such as cross-language and cross-TTS evaluation—improves more slowly and exhibits persistent error gaps. This suggests that scaling primarily enhances the extraction of transferable acoustic cues, but does not fully resolve mismatches induced by linguistic or synthesis variability.

Most notably, our robustness analysis reveals a clear asymmetry between scaling detection accuracy and scaling robustness. While robustness to unstructured perturbations, such as Gaussian noise, improves alongside in-distribution detection performance under clean condition, robustness to structured, semantic-preserving distortions, such as pitch shifting and neural codec artifacts, shows much weaker and often saturating gains. This suggests that increasing model capacity tends to reinforce reliance on dominant training cues, rather than promoting invariance to systematic transformations. In practice, these results caution against treating model scale as a reliable proxy for robustness in real-world audio deepfake detection.

Using the fitted in-distribution scaling law of Whisper-Large, we further explore the data requirements needed to reach low-error regimes under hypothetical improvements to the asymptotic error floor. While the fitted model predicts an irreducible error floor around 0.07 under the current fine-tuning setup, we consider counterfactual scenarios in which this floor is reduced through improved architectures, supervision, or data quality. Under a conservative assumption of a 3% error floor, the scaling extrapolation suggests that achieving 5% EER would require approximately 24 million training samples, while an idealized zero-floor scenario yields an estimate of roughly 7 million samples. Although these projections rely on extrapolation beyond the observed data range and should be interpreted cautiously, they provide a quantitative sense of the data scale required to reach low-error regimes and underscore the substantial data demands of further performance gains.

**Limitations.** Our work has several limitations. First, although we study scaling over more than an order of magnitude in model size, all experiments are conducted within a single model family (Whisper). While this control is essential for isolating scaling effects, extending the analysis to other architectures or pretraining paradigms may reveal different scaling regimes. Second, our scaling analysis focuses on post-training fine-tuning and does not address how scaling laws interact with large-scale pretraining or joint pretraining–fine-tuning strategies. As pretraining datasets and objectives evolve, the resulting scaling behavior in downstream detection may change. Additionally, given the limited number of discrete model sizes, our fitted scaling curves should be interpreted qualitatively rather than as precise estimates of scaling exponents, especially in robustness and cross-domain settings where performance saturates.

## 7. Conclusion

We presented a systematic study of post-training scaling in audio deepfake detection, analyzing how detection performance (EER), robustness, and generalization to unseen languages and TTS systems evolve with detector capacity and dataset size under controlled pretraining and architecture. By focusing on fine-tuning regimes, our analysis isolates the practical effects of scaling under realistic deployment conditions. We find that increasing model capacity reliably improves in-distribution detection accuracy and generalization to unseen languages and synthesis methods. However, these gains consistently weaken under distribution shifts, with persistent performance gaps remaining even at larger scales. Moreover, robustness does not scale uniformly: while resistance to unstructured perturbations such as Gaussian noise improves with capacity, robustness to structured, semantic-preserving distortions, including pitch manipulation and neural codec artifacts, shows limited or saturating improvements.

Overall, our study demonstrates that scaling is effective yet constrained in audio deepfake detection. While larger models offer clear benefits in accuracy and generalization, scaling alone is insufficient to ensure robustness under realistic conditions. These findings highlight the need to complement model scaling with robustness-aware training strategies, greater data diversity, and targeted augmentations to achieve reliable real-world detection.

## Acknowledgements

The authors thank the partial support from the Fordham–IBM Research Fellowship, the Fordham Faculty Research Grant, and the New York Public Interest Technology Regional Network Seed Grant.

## Impact Statement

This work aims to improve the understanding and design of audio deepfake detection systems by characterizing how detection performance, robustness, and generalization scale with model capacity and data. Our findings provide practical guidance for deploying detectors more reliably under realistic conditions. There are many potential societal consequences of our work, none which we feel must be specifically highlighted here.

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
