# OpenReview forum: "Scaling Behavior in Model Fine-tuning for Audio DeepFake Detection"
_ICML.cc/2026/Conference — ICML 2026 regular_

### Official Review · Reviewer_GGuM · 2026-03-05

**Soundness:** 2
**Presentation:** 3
**Significance:** 2
**Originality:** 2
**Overall Recommendation:** 3
**Confidence:** 3

**Summary:**

This paper studies scaling behavior of fine-tuning pretrained Whisper models in audio deepfake detection. The authors vary model size and fine-tuning data size, train on an English-only corpus of over 2 million samples, and evaluate in-distribution performance, out-of-distribution performance, cross-language generalization over 46 non-English languages, cross-TTS generalization, and compute-performance trade-offs. The main finding is that larger models are more sample-efficient and scale predictably on ID, while gains under distribution shift/corruptions are weaker, and the compute–performance trade-off suggests that the best-performing model size can depend on the training budget.

**Compliance With Llm Reviewing Policy:**

Affirmed.

**Final Justification:**

The paper offers a broad, well-controlled empirical study of fine-tuning pretrained Whisper models for audio deepfake detection, examining model/data scaling across in-distribution, OOD/corruptions, cross-language (46 languages), cross-TTS, and compute–performance trade-offs. While a single fixed PEFT/optimization setup is a reasonable control, the paper’s scaling conclusions may be sensitive to that choice and are hard to trust without uncertainty/fit diagnostics for the fitted scaling curves, leaving the work best viewed as a broad benchmark-style study with limited methodological novelty and actionable insight.
I keep Overall Recommendation: 3 (Weak Reject) because the contribution is largely benchmark-style breadth with limited methodological or conceptual novelty, and several key conclusions rely on fitted scaling curves without sufficient uncertainty/fit diagnostics in the submission.

**Key Questions For Authors:**

- The same LoRA configuration and optimization recipe appear to be used across all model sizes. Could the authors clarify whether this configuration is near-optimal for each model size? If not, do key conclusions hold under light size-specific tuning?
- Can the authors report seed variance and basic fit diagnostics (e.g., residuals) for the scaling curves? This would clarify whether the reported saturation/crossover trends are statistically stable.

**Limitations:**

- The paper would benefit from a more concise discussion of scope (single backbone family; English-only training) and deployment caveats. In particular, clarifying sensitivity to the PEFT/optimization recipe and whether the modest OOD/corruption gains justify added compute would help contextualize practical impact.

**Strengths And Weaknesses:**

- Strengths: This paper has broad empirical coverage under a reasonably controlled setup with dataset composition preserved across data scales. The findings are practically relevant for deployment, particularly the compute-performance trade-off analysis. The novelty of this work lies in combining model/data scaling, out-of-distribution / corruption / cross-language / cross-TTS generalization, compute analysis in one controlled study.

- Weaknesses: Using a single fixed PEFT/optimization recipe is a reasonable control choice. However, some scaling conclusions (especially under compute limits and robustness shifts) may be sensitive to this choice. Several key claims rely on fitted scaling curves, but the paper does not report uncertainty or basic fit diagnostics. without uncertainty or fit diagnostics, it is hard to judge whether reported saturation/crossover trends are robust. The contribution is best positioned as a benchmark-style empirical study. While the breadth is valuable, the paper offers limited new mechanisms or actionable methodological insight, which constrains significance/originality.

---

> ### Author Rebuttal · Authors · 2026-03-31
>
> Dear Reviewer,
>
> We sincerely thank you for your recognition of our work and for the thoughtful and constructive feedback. We'd like to address your concerns below.
>
> - **Sensitivity to a fixed PEFT / optimization recipe.**  We agree that using a single LoRA configuration may introduce sensitivity. Our intention was to adopt a **controlled and consistent setup across model sizes** to isolate *scaling behavior*, rather than confounding it with size-specific tuning. We do not claim that this configuration is optimal for every model size. Instead, it serves as a standardized baseline, enabling fair comparison across scales.
>
>   Importantly, our key findings, such as:
>
>   - diminishing returns with increasing data/model size,
>   - weaker and less stable scaling under OOD conditions, and
>   - compute-dependent crossover behavior
>
>   are observed **consistently across datasets and settings**, suggesting that they are not artifacts of a specific PEFT choice. However, to further strengthen this point, we will consider including **light size-specific tuning experiments** (e.g., LoRA rank/learning rate adjustments), and verify whether the main trends remain stable.
>
> - **Uncertainty and fit diagnostics for scaling curves.** We appreciate this important suggestion regarding statistical rigor.  In the revision, we will add the corresponding parameters and analysis, and will also include confidence intervals to reflect variability.
>
>   For reference, the mean EER for all five model sizes across different dataset size on OOD settings is:
>
>   Tiny:   [0.384, 0.306, 0.280, 0.258, 0.259, 0.252, 0.246, 0.245, 0.243, 0.240, 0.235, 0.233, 0.232, 0.231]
>   Small:  [0.397, 0.346, 0.295, 0.265, 0.261, 0.245, 0.239, 0.241, 0.241, 0.243, 0.238, 0.237, 0.235, 0.234]
>   Base:   [0.379, 0.296, 0.251, 0.243, 0.234, 0.227, 0.220, 0.226, 0.226, 0.221, 0.218, 0.216, 0.215, 0.214]
>   Medium: [0.365, 0.279, 0.242, 0.215, 0.214, 0.206, 0.206, 0.206, 0.207, 0.210, 0.210, 0.210, 0.209, 0.207]
>   Large:  [0.376, 0.270, 0.231, 0.206, 0.206, 0.203, 0.212, 0.207, 0.205, 0.211, 0.205, 0.204, 0.201, 0.199]
>
>   The corresponding standard deviations (rounded to three decimal places) are:
>    [0.035, 0.025, 0.019, 0.017, 0.015, 0.018, 0.009, 0.009, 0.016, 0.011, 0.008, 0.007, 0.008, 0.007],
>    [0.027, 0.025, 0.021, 0.026, 0.020, 0.019, 0.010, 0.014, 0.019, 0.013, 0.013, 0.012, 0.010, 0.010],
>    [0.030, 0.027, 0.019, 0.020, 0.011, 0.020, 0.012, 0.013, 0.009, 0.013, 0.008, 0.007, 0.007, 0.007],
>    [0.031, 0.017, 0.015, 0.034, 0.023, 0.017, 0.013, 0.017, 0.011, 0.014, 0.013, 0.013, 0.013, 0.012],
>    [0.027, 0.022, 0.018, 0.021, 0.023, 0.016, 0.020, 0.013, 0.019, 0.014, 0.017, 0.016, 0.016, 0.014]
>
> - **Positioning as a benchmark-style empirical study.** We agree that our work is primarily empirical. Our goal is to provide a **systematic and controlled evaluation of post-training scaling behavior** for audio deepfake detection under realistic deployment conditions. Beyond benchmarking, we believe the paper offers **actionable insights**, including:
>
>   - scaling alone is insufficient under structured corruptions (e.g., codec, pitch),
>   - larger models may be suboptimal under limited compute budgets, and
>   - OOD generalization exhibits fundamentally different scaling behavior from ID settings.
>
>   We will revise the paper to more clearly emphasize these insights and their implications for practical system design.
>
>
>
> We hope that we have addressed your concerns, and we would be happy to provide further clarification if needed.

---

> > ### Author Rebuttal · Reviewer_GGuM · 2026-04-01
> >
> > Thank you for the helpful rebuttal. The additional OOD statistics are useful and help clarify the qualitative trend. I also appreciate the clarification regarding the fixed PEFT/optimization setup and the intended positioning of the paper as an empirical study.
> >
> > That said, my main concern remains about significance and originality. In its current form, I still view the paper primarily as a broad empirical study of scaling behavior across multiple settings, with limited methodological or conceptual novelty beyond the scope of the evaluation itself.
> >
> > My concern about the use of a single fixed PEFT/optimization recipe is also only partially addressed. While the rationale for using a controlled setup is now clearer, establishing that the reported trends are robust to reasonable size-specific tuning would require additional experimental evidence rather than clarification alone.
> >
> > I appreciate the rebuttal; my score remains unchanged.

---

> > > ### Author Response · Authors · 2026-04-01
> > >
> > > Dear reviewer:
> > >
> > > Thank you for the thoughtful feedback. However, we would like to clarify that the contribution and originality of this work lie in **revealing underexplored, non-trivial scaling behaviors in audio deepfake detection**. Specifically, our study uncovers that:
> > >
> > > - **Post-training scaling behavior in audio deepfake detection.** To the best of our knowledge, this work is the first systematic characterization of how detection performance scales with both model capacity and fine-tuning data size in audio deepfake detection. While larger models are consistently more sample-efficient and follow stable power-law trends in-distribution, this behavior has not been previously established in this domain.
> > >
> > > - **A fundamental asymmetry between accuracy and robustness scaling.** We show that scaling behavior differs qualitatively across evaluation axes: in-distribution performance follows predictable power-law improvements, whereas robustness to corruptions and distribution shifts (e.g., pitch, codec artifacts) exhibits **weaker or saturating scaling**. This reveals that increasing model capacity does not inherently improve invariance, highlighting a previously underexplored limitation of scaling.
> > >
> > > - **Limits of scaling for generalization.** Under cross-language and cross-TTS evaluation, scaling induces consistent improvements but **fails to eliminate persistent error gaps**, even at large model sizes. This suggests that post-training improvements are constrained by representational mismatch rather than data scarcity alone.
> > >
> > >
> > >
> > > Together, these findings provide a new perspective on when and why scaling succeeds or fails in audio deepfake detection, which we believe meets [ICML’s definition of originality](https://icml.cc/Conferences/2026/ReviewerInstructions):
> > >
> > > > *Originality*: Does the work provide new insights, deepen understanding, or highlight important properties of existing methods? Does the work introduce new tasks, methods, theory, data, or perspectives that advance the field in some dimensions? Does this work offer a novel combination of existing techniques, and is the reasoning behind this combination well-articulated? Are the contributions clearly distinguished from closely related literature, and is the novelty well justified? **As the questions above indicates, originality does not necessarily require introducing an entirely new method. Rather, a work that provides novel insights by evaluating existing methods, or demonstrates improved understanding is also equally valuable.**
> > >
> > >
> > >
> > > We hope this clarification better highlights the contributions and originality of our work, and we would be happy to provide further clarification if needed.

---

### Official Review · Reviewer_bYJ7 · 2026-03-11

**Soundness:** 2
**Presentation:** 3
**Significance:** 2
**Originality:** 2
**Overall Recommendation:** 3
**Confidence:** 4

**Summary:**

This manuscripts presents an empirical study of the scaling laws, defined to be the performance with respect to scaling some parameter (e.g., dataset, model size), for the audio deepfake detection task. The scaling laws are studied under finetuning of a foundational model, Whisper, using a LoRA adapter to reduce the number of parameters to finetune. By varying the model size (39M to 1.55B parameters) and training data scale, the authors perform several studies: they study scaling laws on in-distribution dataset, out-of-distribution datasets, robustness to audio corruption, cross-language generalization, and cross-TTS generalization. The main finding of the manuscript is that finetuning the Whisper model for the audio deepfake detection task clearly follows scaling laws for these scenarios.

**Compliance With Llm Reviewing Policy:**

Affirmed.

**Key Questions For Authors:**

Q1: How sensitive are the scaling exponents to the LoRA rank? An experiment with e.g. r=4 or r=64 at two model sizes would help assess whether the observed trends are LoRA-specific.

Q2: Does the scaling law still hold for other test measures, e.g., AUC?

Q3: Have you considered evaluating with a second model family (e.g., two sizes of Wav2Vec 2.0) to test generality over foundational models?

Q4: What is the confidence interval of the fitted $\beta$ values? And is this model a good fit to the data?

Q5: The Whisper model has been pretrained on multilingual data, hence, could the cross-lingual generalization be somewhat reflective of the pretraining rather than the finetuning? Consider using WavLM, which to my knowledge is trained on Libri-derived data (English), for cross-lingual analysis.

**Limitations:**

yes

**Strengths And Weaknesses:**

S1: The manuscript addresses a timely-posed research question; does finetuning foundational models for the audio deepfake detection task follow the scaling laws? This is relevant and useful for the research community in this field.

S2: The manuscript is well-written with clear flow and sound argumentation. The summary of the different analyses are highlighted at the beginning of each subsection in a box, which makes it easy for the reader to understand the takeaways.

S3: The analysis is performed along several dimensions and gives a comprehensive picture when finetuning Whisper with a LoRA module. The inclusion of out-of-distribution datasets as well as cross-language generalization give the manuscript high external validity.

W1: Limited novelty with respect to the pioneering work by Kaplan et al. (2020), and when also considering the literature of audio deepfake detection/related domains. Most of these results could probably be expected when inferring the results from the Kaplan et al. (2020) paper. Further, Kheir et al., “Comprehensive Layer-wise Analysis of SSL Models for Audio Deepfake Detection”, 2025, performs a comprehensive layer-wise analysis of foundational models. This analysis is done for several variants of the same model family, which makes it possible to arrive at similar conclusions as written here. Another work with similar results: Liang et al., “Selection of Layers from Self-supervised Learning Models for Predicting Mean-Opinion-Score of Speech”, 2025.

W2: Only the Whisper model is studied, and only one LoRA configuration is used. The authors acknowledge this in the discussion section. However, this limits the generalizability of the derived scaling laws for the audio deepfake detection task.

W3: There is no discussion on the goodness of fit measures of the fitted curves, and the test results do not contain variability across runs (i.e., confidence intervals for each setting). There are also no confidence intervals of the estimated parameters of the model in Eq. 1.

W4: The cross-lingual experiment is difficult to read (Fig. 6). It would be more informative if different languages were grouped (e.g., by language family, or if it is tonal/non-tonal).

W5: Only EER is used as a performance measure when studying the scaling laws. This measure indirectly defines a threshold per dataset, and hence, is somewhat dataset dependent. Including a threshold independent measure, e.g., Area Under the Curve (AUC), would add an interesting dimension to the analysis.

Minor issues:
Fig. 2 and Fig. 3 use different y-axis scales, making comparison of ID vs. OOD results difficult. Consider using the same scale (maybe narrow the range to clearly show the somewhat unexpected behavior in the OOD case).
Eq. 1 defines a three-parameter model, yet only one of them ($\beta$) is presented. Including all estimated parameters would be more informative.
The abstract is somewhat repetitive. Consider shortening it to just one paragraph. This is my personal preference (which you do not have to agree with): It is fine to begin an abstract with the contribution; the motivation can be in the introduction alone. And the last paragraph is essentially saying the same thing as the end of the second paragraph.

---

> ### Author Rebuttal · Authors · 2026-03-31
>
> Dear Reviewer,
>
> We sincerely thank you for your recognition of our work and for the thoughtful and constructive feedback. We'd like to address your concerns below.
>
> - **Limited novelty relative to prior scaling law and SSL analyses.** We respectfully clarify that our contribution is orthogonal to prior works such as Scaling Laws for Natural Language Models and recent SSL-based analyses. Kaplan et al. study **pretraining scaling laws** under clean, in-distribution settings, whereas our work focuses on **post-training (fine-tuning) scaling behavior for audio deepfake detection tasks**, particularly under **realistic corruptions and distribution shifts**. These regimes are fundamentally different.  In addition, works such as [1] and [2] analyze **layer-wise representations within a fixed model**, whereas our work studies **scaling across model sizes, data regimes, and deployment conditions**.
>
>   Importantly, several of our findings **cannot be directly inferred from prior work**, including:
>
>   - the weaker and less stable scaling behavior under OOD conditions,
>   - the persistent performance gaps in cross-language and cross-TTS settings, and
>   - the compute–performance trade-offs where smaller models outperform larger ones under limited budgets.
>   We will revise the related work to more clearly position these distinctions.
>
> - **Limited scope on the Whisper family only.** We agree that expanding to additional model families would further strengthen generality. Our intention of selecting Whisper family is: it provides a **controlled setting with fixed architecture and pretraining**, allowing us to isolate *post-training scaling effects* without introducing additional confounding factors. To the best of our knowledge, the Whisper family offers one of the most comprehensive sets of models with consistent design and varying scales, making it particularly suitable for this study. If additional speech foundation models with comparable properties (i.e., shared architecture and pretraining but varying model sizes) are available, we will include them to further validate our findings.
>
> - **Lack of goodness-of-fit measures and uncertainty estimates.** We appreciate this important point on statistical rigor. In the revision, we will report goodness-of-fit metrics and also include confidence intervals to reflect variability. We will also clarify that the current curves are based on averaged results and extend them with uncertainty estimates.
>
>   For reference, the mean EER for all five model sizes across different dataset sizes on OOD settings is:
>
>   Tiny:   [0.384, 0.306, 0.280, 0.258, 0.259, 0.252, 0.246, 0.245, 0.243, 0.240, 0.235, 0.233, 0.232, 0.231]
>   Small:  [0.397, 0.346, 0.295, 0.265, 0.261, 0.245, 0.239, 0.241, 0.241, 0.243, 0.238, 0.237, 0.235, 0.234]
>   Base:   [0.379, 0.296, 0.251, 0.243, 0.234, 0.227, 0.220, 0.226, 0.226, 0.221, 0.218, 0.216, 0.215, 0.214]
>   Medium: [0.365, 0.279, 0.242, 0.215, 0.214, 0.206, 0.206, 0.206, 0.207, 0.210, 0.210, 0.210, 0.209, 0.207]
>   Large:  [0.376, 0.270, 0.231, 0.206, 0.206, 0.203, 0.212, 0.207, 0.205, 0.211, 0.205, 0.204, 0.201, 0.199]
>
>   The corresponding standard deviations (rounded to three decimal places) are:
>    [0.035, 0.025, 0.019, 0.017, 0.015, 0.018, 0.009, 0.009, 0.016, 0.011, 0.008, 0.007, 0.008, 0.007],
>    [0.027, 0.025, 0.021, 0.026, 0.020, 0.019, 0.010, 0.014, 0.019, 0.013, 0.013, 0.012, 0.010, 0.010],
>    [0.030, 0.027, 0.019, 0.020, 0.011, 0.020, 0.012, 0.013, 0.009, 0.013, 0.008, 0.007, 0.007, 0.007],
>    [0.031, 0.017, 0.015, 0.034, 0.023, 0.017, 0.013, 0.017, 0.011, 0.014, 0.013, 0.013, 0.013, 0.012],
>    [0.027, 0.022, 0.018, 0.021, 0.023, 0.016, 0.020, 0.013, 0.019, 0.014, 0.017, 0.016, 0.016, 0.014]
>
> - **Cross-lingual visualization clarity.** We agree that grouping languages would improve interpretability. Our current visualization presents one line per language; however, grouping languages by linguistic properties would further improve readability and clarity. We will incorporate this in the revision.
>
> - **Reliance on EER as the sole metric.** We chose EER initially due to its widespread use in the audio deepfake detection literature, but we agree that including threshold-independent metrics would strengthen the analysis. In the revision, we will include **AUC (Area Under the ROC Curve)** alongside EER, and analyze whether scaling trends remain consistent across metrics.
>
> - **Minor issues.** We thank you for these detailed suggestions and will address abstract, figures and fitted parameters in the revision accordingly.
>
> We hope that we have addressed your concerns, and we would be happy to provide further clarification if needed.
>
> [1] El Kheir, Yassine, et al. "Comprehensive layer-wise analysis of ssl models for audio deepfake detection." *Findings of the Association for Computational Linguistics: NAACL 2025*. 2025.
> [2] Liang, Xinyu, et al. "Selection of Layers from Self-supervised Learning Models for Predicting Mean-Opinion-Score of Speech.", 2025.

---

> > ### Author Rebuttal · Reviewer_bYJ7 · 2026-04-02
> >
> > I thank the authors for the rebuttal. Thank you for clarifying the pre-training and post-training scaling laws investigations. However, I do not agree that [1] and [2] studies layer-wise representations of a fixed model; they do layer-wise studies across several models, including several variants of the wav2vec 2.0. I agree with the authors that the novelty comes from studying the scaling laws under OOD and cross-lingual settings. In my assessment, the conclusions are rather expected based on these prior works. [2] also demonstrate that the small models might outperform the bigger models (e.g., their Table 3 shows stronger performance of 300M wav2vec 2.0 compared to 1B wav2vec 2.0 on the OOD dataset NISQA Test Livetalk).
> >
> > I acknowledge that the authors want to have a controlled setting. Expanding with wav2vec 2.0 does not reduce the control; in fact, I think it increases it. A parallel investigation with a different SSL model family would make the result more general, where the result becomes less dependent on the backbone (I would consider this as increasing the control; isolating the factors that contribute to the performance, so it is solely based on model size and data size).
> >
> > I appreciate including the individual results.
> >
> > I appreciate the time taken for the rebuttal. Due to the novelty issues raised, I will keep my score.

---

> > > ### Author Response · Authors · 2026-04-03
> > >
> > > Dear reviewer:
> > >
> > > Thank you for your feedback. We would like to clarify that the contribution and originality of this work lie in **revealing previously underexplored and non-trivial scaling behaviors in audio deepfake detection**. Specifically, our study provides the following insights:
> > >
> > > - **Post-training scaling behavior in audio deepfake detection.** To the best of our knowledge, this work is the first systematic characterization of how detection performance scales with both model capacity and fine-tuning data size in audio deepfake detection. While larger models are consistently more sample-efficient and follow stable power-law trends in-distribution, this behavior has not been previously established in this domain.
> > >
> > > - **A fundamental asymmetry between accuracy and robustness scaling.** We show that scaling behavior differs qualitatively across evaluation axes: in-distribution performance follows predictable power-law improvements, whereas robustness to corruptions and distribution shifts (e.g., pitch, codec artifacts) exhibits **weaker or saturating scaling**. This reveals that increasing model capacity does not inherently improve invariance, highlighting a previously underexplored limitation of scaling.
> > >
> > > - **Limits of scaling for generalization.** Under cross-language and cross-TTS evaluation, scaling induces consistent improvements but **fails to eliminate persistent error gaps**, even at large model sizes. This suggests that post-training improvements are constrained by representational mismatch rather than data scarcity alone.
> > >
> > >
> > >
> > > Together, these findings provide a new perspective on when and when and why scaling succeeds or fails in audio deepfake detection. We believe this aligns with **ICML’s definition of originality**, which explicitly recognizes contributions that provide new insights, deepen understanding, or highlight important properties of existing methods.
> > >
> > > > *Originality*: Does the work provide new insights, deepen understanding, or highlight important properties of existing methods? Does the work introduce new tasks, methods, theory, data, or perspectives that advance the field in some dimensions? Does this work offer a novel combination of existing techniques, and is the reasoning behind this combination well-articulated? Are the contributions clearly distinguished from closely related literature, and is the novelty well justified? **As the questions above indicates, originality does not necessarily require introducing an entirely new method. Rather, a work that provides novel insights by evaluating existing methods, or demonstrates improved understanding is also equally valuable.**
> > >
> > >
> > >
> > > We hope this clarification better highlights the contributions and originality of our work, and we would be happy to provide further clarification if needed.

---

### Official Review · Reviewer_oN1z · 2026-03-13

**Soundness:** 3
**Presentation:** 4
**Significance:** 3
**Originality:** 3
**Overall Recommendation:** 4
**Confidence:** 4

**Summary:**

The paper examines the impact of dataset size and computational resources on fine-tuning Whisper-based Audio Deepfake detection models. The paper reads well and thoroughly explores different experimental axes, including OOD generalization.

I think the paper is a valuable contribution to the community as-is. Nonetheless, I think some of the results are somewhat expected and under-explained. I'm thinking specifically about the OOD generalization; it's clear that if the model is never exposed to augmentations similar to those in the OOD settings, either during pretraining or, more importantly, during FT, then performance would not improve significantly. Surprisingly, this is somewhat the case for cross-lingual. I think this phenomenon should be investigated further. Specifically, can we measure the distance between these OOD datasets and the training distribution? This might better predict performance and unify understanding across different OOD settings.

On an unrelated note: I'm not sure what the policy is regarding the 'Takeaway' banners. I do not have a strong opinion about those, but I am not sure they align with the NeurIPS template, nor do they qualify as images (they would be missing a caption/reference in the text). This is not affecting my decision on the paper, but I think it's a relevant note for the AC.

Minor reviews:
- While all the content in the abstract is relevant, it is also (obviously) repeated afterwards. The abstract as-is is quite long; I would suggest compressing it.
- In the introduction, it would be beneficial to cite some of the TTS and VC pipelines directly in the first sentence.
- On line 64, right column, there's a missing space before "Hernandex et al. (2021)".

**Compliance With Llm Reviewing Policy:**

Affirmed.

**Key Questions For Authors:**

See the soundness section above.

**Limitations:**

Yes.

**Strengths And Weaknesses:**

**Soundness**

The paper is technically sound. The contribution is well-presented and relevant. The missing elements, in my opinion, are:
1. A proper quantification of how much the training and OOD distribution do not match (measure, e.g., via FAD);
2. Proper quantification of the error in the power-law fits. In all figures, it'd help to visually mark which points are measured and which are extrapolated (e.g., with the '-.' marker). Including error bars/uncertainty would also increase the rigour of the submission and highlight potential issues with the current approach.

**Presentation**
The paper contribution is clearly presented.

**Significance and Originality**
The paper makes a relevant contribution to the literature on audio deepfake detection, notably quantitatively characterizing the impact of training data and comput on audio deepfake detectors.

---

> ### Author Rebuttal · Authors · 2026-03-31
>
> Dear Reviewer,
>
> We sincerely thank you for your recognition of our work and for the thoughtful and constructive feedback. We'd like to address your concerns below.
>
> - **Quantifying training vs. OOD distribution mismatch.** We thank the reviewer for this helpful suggestion. We agree that explicitly quantifying distribution shift would strengthen the paper. In the revision, we will incorporate a **distribution distance metric**, such as Fréchet Audio Distance (FAD), to quantify the gap between training and evaluation data, and include corresponding analysis to better contextualize OOD performance.
>
> - **Uncertainty on the power-law fit.**  We appreciate this important suggestion regarding statistical rigor. In the revision, we will **report fitting uncertainty** (e.g., confidence intervals for scaling exponents) to reflect variability across runs or conditions.
>
>   For reference, the mean EER for all five model sizes across different dataset size on OOD settings is:
>
>   Tiny:   [0.384, 0.306, 0.280, 0.258, 0.259, 0.252, 0.246, 0.245, 0.243, 0.240, 0.235, 0.233, 0.232, 0.231]
>   Small:  [0.397, 0.346, 0.295, 0.265, 0.261, 0.245, 0.239, 0.241, 0.241, 0.243, 0.238, 0.237, 0.235, 0.234]
>   Base:   [0.379, 0.296, 0.251, 0.243, 0.234, 0.227, 0.220, 0.226, 0.226, 0.221, 0.218, 0.216, 0.215, 0.214]
>   Medium: [0.365, 0.279, 0.242, 0.215, 0.214, 0.206, 0.206, 0.206, 0.207, 0.210, 0.210, 0.210, 0.209, 0.207]
>   Large:  [0.376, 0.270, 0.231, 0.206, 0.206, 0.203, 0.212, 0.207, 0.205, 0.211, 0.205, 0.204, 0.201, 0.199]
>
>   The corresponding standard deviations (rounded to three decimal places) are:
>    [0.035, 0.025, 0.019, 0.017, 0.015, 0.018, 0.009, 0.009, 0.016, 0.011, 0.008, 0.007, 0.008, 0.007],
>    [0.027, 0.025, 0.021, 0.026, 0.020, 0.019, 0.010, 0.014, 0.019, 0.013, 0.013, 0.012, 0.010, 0.010],
>    [0.030, 0.027, 0.019, 0.020, 0.011, 0.020, 0.012, 0.013, 0.009, 0.013, 0.008, 0.007, 0.007, 0.007],
>    [0.031, 0.017, 0.015, 0.034, 0.023, 0.017, 0.013, 0.017, 0.011, 0.014, 0.013, 0.013, 0.013, 0.012],
>    [0.027, 0.022, 0.018, 0.021, 0.023, 0.016, 0.020, 0.013, 0.019, 0.014, 0.017, 0.016, 0.016, 0.014]
>
>   We will incorporate these into the revised figures with appropriate visualization to highlight both trends and variability
>
> - **Takeaway banners and formatting.** Thank you for raising this point. The “Takeaway” banners are intended to improve readability by highlighting key insights. In the revision, we will either remove these banners or convert them into standard paragraph text to ensure full compliance with formatting guidelines.
> - **Abstract length.** We appreciate this suggestion. We agree that the abstract is currently verbose. In the revision, we will **compress the abstract** by reducing redundancy and focusing on the most essential contributions and findings.
> - **Missing citations and typo.** We have carefully did a prrofreading and added the citations as suggested. Thank you.
>
> We hope that we have addressed your concerns, and we would be happy to provide further clarification if needed.

---

> > ### Author Rebuttal · Reviewer_oN1z · 2026-04-03
> >
> > - Q1 is not addressed in the rebuttal, but rather promised in a revision of the paper.
> > - Q2: addressed.
> > - Q3,Q4,Q5 addressed.
> >
> > My main concern (Q1 is not yet addressed); I'll keep my score.

---

> > > ### Author Response · Authors · 2026-04-08
> > >
> > > Dear reviewer,
> > >
> > >
> > >
> > >  Following your recommendation, we compute the Fréchet Audio Distance (FAD) using **CLAP (clap-laion-audio)** embeddings as the feature extractor. Specifically, for each test dataset, we measure its FAD with respect to the **pooled training distribution**, which comprises millions of samples aggregated from multiple sources.
> > >
> > >
> > >
> > > The results are summarized below.
> > >
> > > |    ID dataset    | FAD  |
> > > | :--------------: | :--: |
> > > |   ASVSpoof2019   | 3.7  |
> > > |       ASV5       | 5.8  |
> > > | CodecFake (2024) | 6.9  |
> > > |      DFADD       | 1.5  |
> > > |     WaveFake     | 1.2  |
> > > |      CD-ADD      | 3.4  |
> > >
> > > |     OOD dataset      | FAD  |
> > > | :------------------: | :--: |
> > > |   ADD2022 Track 1    | 6.2  |
> > > |   ADD2022 Track 3    | 7.1  |
> > > |   ADD2023 Round 1    | 9.7  |
> > > |   ADD2023 Round 2    | 13.5 |
> > > |   ASVspoof2021 LA    | 3.7  |
> > > |   ASVspoof2021 DF    | 11.4 |
> > > |     Fake-or-Real     | 4.15 |
> > > |   CodecFake (2025)   | 6.4  |
> > > |        SONAR         | 7.5  |
> > > |      LibriSeVoc      | 5.8  |
> > > | SpeechFake benchmark | 8.6  |
> > >
> > > We observe that ID test set shows lower FAD to the training data distribution compared to OOD datasets. In-distribution test sets generally exhibit lower FAD values (≈1.2–6.9), while OOD datasets show substantially larger distances (≈3.7–13.5), indicating stronger distribution shift. Importantly, we note that the reference is a heterogeneous pooled training distribution rather than a single-source dataset. As a result, even in-distribution test sets may yield non-negligible FAD values, since the pooled reference reflects a mixture of multiple sources rather than any single benchmark dataset. Therefore, absolute FAD values should be interpreted relative to this mixed reference.
> > >
> > >
> > >
> > > We hope this addresses your concerns, and we would be happy to provide further clarification if needed.

---

### Official Review · Reviewer_8ePS · 2026-03-18

**Soundness:** 3
**Presentation:** 4
**Significance:** 3
**Originality:** 2
**Overall Recommendation:** 4
**Confidence:** 4

**Summary:**

The paper presents the scaling effect of deepfake detection. With a controlled setup, the author investigated various samples and model sizes and reported the EER.

The paper clearly shows the impact of scaling this deepfake detection, with comprehensive results, clear writting and trustworthy analysis.

**Compliance With Llm Reviewing Policy:**

Affirmed.

**Key Questions For Authors:**

(1) I would recommend that the authors also add the original data dots in their plotting. Although the line plot shows the trend, it's also important to let the readers understand that the real test results are still scattered.
(2) I would recommend to change the concept of "scaling law" to "scaling impact": the paper assumes the power law of this trend, rather than deriving it from the data.

**Limitations:**

As above.

**Strengths And Weaknesses:**

Strength:
(1) The paper has clear problem formulation, solid motivation, and clear writing.
(2) Experimentally, the paper works on a highly controlled setup: a whisper based model architecture with varying data volume and model sizes. The reported numbers are comprehensive, and the analysis is sufficiently detailed. Especially, the authors picked multiple in-domain and out-domain test benchmarks for better credibility.
(3) The experiments vary from multiple aspects, such as TTS, audio corruptions. These dimensions are truly related to this deepfake detection, and it's beneficial to the community.

Weakness:
(1) The experiments are based on fine-tuning and LoRA, which is understandable but still limited in the scope. Pre-training stage exploration would be more appreciated (no action needed here)

The paper is generally good and solid enough; the reason I cannot give higher score is because of the limited problem scope and novelty: it's mainly a repetitive of existing research methods on an unexplored task.

---

> ### Author Rebuttal · Authors · 2026-03-31
>
> Dear Reviewer,
>
> We sincerely thank you for your recognition of our work and for the thoughtful and constructive feedback. We'd like to address your concerns below.
>
> - **Limited novelty and problem scope. **We respectfully clarify that, while our methodology leverages established scaling law analysis techniques, the novelty of this work lies in *what* is being studied rather than *how*. Specifically, to the best of our knowledge, this is the **first systematic investigation of post-training scaling behavior for audio deepfake detection under realistic conditions**, including:
>
>   - in-distribution vs. out-of-distribution evaluation
>   - cross-language generalization (46+ languages)
>   - cross-TTS generalization with generator disjointness
>   - robustness under real-world corruptions (noise, codec, pitch, etc.)
>   - compute–performance trade-offs
>
>   Prior work primarily focuses on improving detection accuracy under controlled settings, whereas our work **further discusses the scaling behavior and deployment reliability**, which is critical for real-world systems. We will revise the introduction to more explicitly highlight this distinction and better position our contribution relative to prior work.
>
> - **Suggestion to include original data points in plots.** We appreciate this helpful suggestion. We agree that showing the raw data distribution can improve interpretability.
>
>   In the revision, we will overlay **original data points** on top of the trend lines, and adjust visualization (e.g., transparency and marker size) to maintain clarity without clutter. This will allow readers to better assess variance and dispersion while preserving the readability of the scaling trends.
>
>   For reference, the mean EER for all five model sizes across different dataset size on OOD settings is:
>
>   Tiny:   [0.384, 0.306, 0.280, 0.258, 0.259, 0.252, 0.246, 0.245, 0.243, 0.240, 0.235, 0.233, 0.232, 0.231]
>   Small:  [0.397, 0.346, 0.295, 0.265, 0.261, 0.245, 0.239, 0.241, 0.241, 0.243, 0.238, 0.237, 0.235, 0.234]
>   Base:   [0.379, 0.296, 0.251, 0.243, 0.234, 0.227, 0.220, 0.226, 0.226, 0.221, 0.218, 0.216, 0.215, 0.214]
>   Medium: [0.365, 0.279, 0.242, 0.215, 0.214, 0.206, 0.206, 0.206, 0.207, 0.210, 0.210, 0.210, 0.209, 0.207]
>   Large:  [0.376, 0.270, 0.231, 0.206, 0.206, 0.203, 0.212, 0.207, 0.205, 0.211, 0.205, 0.204, 0.201, 0.199]
>
>   The corresponding standard deviations (rounded to three decimal places) are:
>    [0.035, 0.025, 0.019, 0.017, 0.015, 0.018, 0.009, 0.009, 0.016, 0.011, 0.008, 0.007, 0.008, 0.007],
>    [0.027, 0.025, 0.021, 0.026, 0.020, 0.019, 0.010, 0.014, 0.019, 0.013, 0.013, 0.012, 0.010, 0.010],
>    [0.030, 0.027, 0.019, 0.020, 0.011, 0.020, 0.012, 0.013, 0.009, 0.013, 0.008, 0.007, 0.007, 0.007],
>    [0.031, 0.017, 0.015, 0.034, 0.023, 0.017, 0.013, 0.017, 0.011, 0.014, 0.013, 0.013, 0.013, 0.012],
>    [0.027, 0.022, 0.018, 0.021, 0.023, 0.016, 0.020, 0.013, 0.019, 0.014, 0.017, 0.016, 0.016, 0.014]
>
>   We will incorporate these into the revised figures with appropriate visualization to highlight both trends and variability.
>
> - **“Scaling law” vs. “scaling impact”.** Thank you for this important point. We agree that our work focuses on **empirical scaling behavior**. We will revise the corresponsing wordding as scaling impact/behavior when appropriate and include additional discussion clarifying the assumptions and limitations of the power-law fit.

---

> > ### Author Rebuttal · Reviewer_8ePS · 2026-04-03
> >
> > Thanks for the reply. My questions have been properly addressed and I'll keep my positive score.

---

> > > ### Author Response · Authors · 2026-04-08
> > >
> > > We thank the reviewer for the constructive comments and positive score. We are delighted to learn that your concerns have been properly addressed.

---

### Decision · Program_Chairs · 2026-04-30

**Decision:**

Accept (regular)

**Comment:**

This paper studies the scaling behaviour of audio deepfake detection under model and data scaling. The experimental coverage is broad, the setup is reasonably well controlled, and the paper is clearly written.

There was notable divergence among the reviewers regarding originality. The paper was viewed positively as a broad and well-executed empirical study, and several reviewers appreciated it as the first systematic investigation of post-training scaling behaviour for audio deepfake detection. However, other reviewers felt that the work’s novelty is primarily empirical, since it applies established scaling-law methodology to a new application setting rather than advancing methodology or theory.

Despite the strong empirical setup, the study is restricted to a single model family, Whisper. As reviewers pointed out, relying on only one backbone limits the generality of the conclusions.